

# Serum expression of Vascular Endothelial-Cadherin, CD44, Human High mobility group B1, Kallikrein 6 proteins in different stages of laryngeal intraepithelial lesions and early glottis cancer

Michał Żurek[1,2], Anna Rzepakowska[1], Iwona Kotuła[3], Urszula Demkow[3] and Kazimierz Niemczyk[1]

[1] Department of Otorhinolaryngology Head and Neck Surgery, Medical University of Warsaw, Warsaw, Poland
[2] Doctoral School, Medical University of Warsaw, Warsaw, Poland
[3] Department of Laboratory Diagnostics and Clinical Immunology of Developmental Age, Medical University of Warsaw, Warsaw, Poland

Corresponding author
Anna Rzepakowska,
arzepakowska@wum.edu.pl

## ABSTRACT

**Background:** The study was designed to evaluate the potential validity and utility of selected molecular markers in serum samples from patients with specific stages of laryngeal intraepithelial lesions that could serve as diagnostic tools in differentiation of benign and dysplastic lesions from invasive pathologies.

**Methods:** Prospective study included 80 consecutive patients with vocal fold lesions treated at the single otorhinolaryngology centre. All participants had surgical resection of the lesion. Blood samples were collected from each patient before the surgery. Final diagnosis was confirmed on histopathological examination and included 39 (48.75%) non-dysplastic lesions, eight (10%) low-grade dysplasia, six (7.5%) high-grade dysplasia and 27 (33.75%) invasive cancers. The ELISA procedures were performed according to the manufacturer's instruction. Individual serum concentration of selected proteins was reported in ng/ml: Vascular Endothelial-Cadherin Complex (VE-cad), CD44, Human High mobility group protein B1(HMGB1), Kallikrein 6.

**Results:** The highest mean levels of HMGB1, KLK6 and VE-cad were detected in sera of patients with low-grade dysplasia (81.14, 24.33, 14.17 respectively). Soluble CD44 was the most elevated in patients with non-dysplastic lesions (2.49). The HMGB1, KLK6 and VE-cad serum levels were increasing from non-dysplastic to low-grade dysplasia and followed by the decrease for high-grade dysplasia and invasive cancer, however the differences were not significant (*p*-values 0.897, 0.354, 0.1 respectively). Patients' serum had the highest CD44 concentration in non-dysplastic and low-grade dysplasia with the following decrease through high-grade dysplasia and invasive cancer. GERD symptomatic patients had higher levels of KLK6 and CD44 than other patients (*p*-value 0.06 and 0.084 respectively). There were no significant differences of biomarkers levels related to patients' gender (*p*-value from 0.243 to 1) or smoking status (*p*-value from 0.22 to 0.706).

**Conclusions:** VE-cad, HMGB1, CD44 and KLK6 did not prove to be reliable biomarkers implicating malignant potential within vocal fold hypertrophic intraepithelial lesions.

## INTRODUCTION

Laryngeal squamous cell carcinoma (LSCC) is one of the most common head and neck malignancy (*Almadori et al., 2005*; *Chu & Kim, 2008*; *Lionello, Staffieri & Marioni, 2012*). A total of 210,606 new cases of LSCC have been diagnosed in 2017 worldwide (2.76 new cases per 100,000 inhabitants) (*Nocini et al., 2020*). The recent data of the International Agency of Research on Cancer provided the number of 184,615 new laryngeal cancers in 2020 and the proportion of 22.5% of all cases in Europe (*Globocan, 2020*). Despite the development of diagnostic techniques and therapeutic options, there has been no improvement in the 5-year survival for patients with LSCC over the past three decades (*Lionello, Staffieri & Marioni, 2012*; *Nocini et al., 2020*). As a result, the most significant obstacle to overcome is the detection, accurate diagnosis, and treatment of laryngeal precancerous lesions and cancer in its early stages.

LSCCs develop according to a multistep process characterized by the increasing accumulation of genetic changes within epithelial cells. Specific clinical and histopathological stages are associated with progressive architectural and cytological alterations of the mucosa. Precancerous and malignant vocal fold lesions have similar appearance on clinical investigation, but the diagnosis may range from squamous cell hyperplasia through dysplasia and carcinoma *in situ* to malignant cancer (*Gale et al., 2016*; *Gale, Poljak & Zidar, 2017*). Recently the modern techniques of enhanced endoscopy as Narrow Band Imaging enabled improvement in accuracy of preoperative assessment; however, the biopsy and histopathological diagnosis are still the standard procedures for diagnosis confirmation (*Kim et al., 2020*). The laryngeal epithelial lesions are pathologically classified according to World Health Organization recommendations into four stages: no dysplasia, low-grade dysplasia, high-grade dysplasia (including carcinoma *in situ*) and invasive cancer (*Gale, Poljak & Zidar, 2017*).

The progression of morphologic epithelial changes is associated with alterations in processes controlling proliferation, cell adhesion, angiogenesis, epithelial-mesenchymal transformation and cell invasion. Precancerous lesions have been investigated widely to identify reliable markers of malignant transformation; however to date, no solid and useful one has been recognized and the researches continue. The most extensively investigated proteins were related with proliferation and cell cycle control (*e.g.* proliferating cell nuclear antigen, Ki67, p53, p16, p21, p27, and cyclin D1) (*Rodrigo et al., 2012*). However, the prognostic significance of those potential laryngeal biomarkers occurred controversial regarding the results of studies included into the analysis. Noteworthy from this review was

the observation that for clinical application the presence of a specific marker should predict laryngeal lesion progression and the analysis revealed that cortactin, focal adhesion kinase, osteopontin and CD44v6 accomplished such thesis.

A recent review of 16 studies evaluating biomarkers predicting malignant potential in vocal fold leukoplakia by *Wan et al. (2021)* confirmed three categories of those factors depending on their biological roles: proliferation and cell cycle control, cell adhesion, and invasion. Nevertheless, the prognostic utility of analyzed biomarkers was limited due to the variable methodologies and studies designs. However, the authors of this review implied that currently available evidence suggests several factors of particular interest in the determination of the prognosis: p53, p16, cyclin D1, IL-10, neutrophil-lymphocyte ratio, osteopontin, CD44v6, cadherin associated protein, cortactin (*Wan et al., 2021*). The conclusions of both reviews suggest further large, well-designed, prospective studies to determine the powerful prognostic biomarkers suitable for implementation in routine clinical practice.

Most of the studies that have been performed so far were retrospective ones and evaluated immunohistochemical expression of specific proteins in tissue samples. It would be ideal to identify relevant clinical markers for differentiation of laryngeal intraepithelial lesions preoperatively in serum or saliva that could ultimately guide the therapeutic decisions.

Our present study was designed to evaluate the potential validity and utility of selected molecular markers in serum samples from patients with specific stages of laryngeal intraepithelial lesions. The aim was to identify potential predictive marker that could serve as diagnostic tool in differentiation of benign, dysplastic and invasive pathologies.

When planning the study, we recognized the markers previously applied in head and neck squamous cell neoplasms and excluded those without significant suitability in conducted studies. We extended the search area to include markers that have shown to be effective in other epithelial tumors.

A final panel of four markers has been accepted and applied in the study to identify reliable serum predictors of malignant epithelial laryngeal lesions.

Vascular endothelial-cadherin (VE-cad), the most important transmembrane component of endothelial adherent junctions, is also related to the formation of vessel-like networks to provide adequate blood supply for tumour growth (*Allegrini et al., 2012*; *Blaise, Polena & Vilgrain, 2015*; *Irani & Dehghan, 2018*; *Sulkowska et al., 2006*). Recently, much attention has been paid to the phenomenon of vasculogenic mimicry, of which VE-cadherin is an element. This phenomenon is considered to be one of the causes of the poor response of head and neck cancers (HNC) to antiangiogenic therapy (*Salem & Salo, 2021*; *Tong et al., 2013*). This is one of the reasons why this potential biomarker was chosen as part of the analysis. Additionally, the analyzes of The Cancer Genome Atlas (TCGA) have confirmed significant statistical differences in the average survival time of patients with HNC depending on the level of VE-cad mRNA expression (*National Cancer Institute, 2022*, https://www.cancer.gov/tcga).

Soluble CD44 protein is another cell adhesion molecule and plays an important role in promotion of uncontrolled growth, evasion of apoptosis, angiogenesis, cell motility

and invasion. Soluble CD44 is regulated by a variety of signaling networks. It plays a vital role in the tumorigenesis through its cooperation with ligands such as hyaluronan, matrix metalloproteinases and osteopontin. CD44 is also involved in tumor initiation and therapeutic resistance (*Chen et al., 2014*; *Dasari, Rajendra & Valluru, 2014*; *Hassn Mesrati et al., 2021*; *Irani & Dehghan, 2018*; *Mishra et al., 2019*; *Seyedmajidi et al., 2018*; *Shinohara et al., 2016*). The analysis of TCGA data revealed that HNC has the second highest CD44 gene expression among all analyzed cancer types (*Ludwig et al., 2019*). Therefore, the level of soluble CD44 was analyzed in this study.

High-mobility group box 1 (HMGB1) is acknowledged as a proinflammatory cytokine that creates a chronic inflammation microenvironment, contributing to the development of inflammation-associated cancers, including epithelial malignancies (*Kang et al., 2014*; *Lee et al., 2012*; *Nguyen et al., 2016*; *Paek et al., 2016*; *Wild et al., 2012*; *Wu et al., 2016*). The HMGB1 is overexpressed in some cancers, including HNC. The expression of HMGB1 genes in the tumor tissue is significantly higher than in the healthy tissue (*Liu et al., 2010*; *Mohajertehran et al., 2018*). There are also polymorphisms in HMGB1 genes (*Lin et al., 2017*). The HMGB1 overexpression in head and neck cancers is also revealed by TCGA as higher FPKM (fragments per kilobase of exon per million reads) than in normal tissues (*National Cancer Institute, 2022*, https://www.cancer.gov/tcga). The role of extracellular HMBG1 in tumorigenesis is still unclear, but gene and protein overexpression have been confirmed in few publications so far so we decided to investigate the role of this protein as a potential biomarker.

Kallikrein 6 (KLK6) is a member of tissue kallikrein related peptidases' family overexpressed in solid cancers. The mechanisms underlying KLK-mediated pericellular proteolysis are not yet fully elucidated; however, there are evidences of microenvironmental enhancement of cancer cell functions promoting proliferation, invasion and metastasis (*Drucker et al., 2015*; *Lan et al., 2016*; *Zhang et al., 2014*). KLK6 expression is regulated by many factors, including microRNA-375, the concentration of which is significantly lower in patients with HNC according analyzes of Gene Expression Omnibus, TGCA and other databases (*Cen et al., 2018*; *Jimenez et al., 2015*). This results, on average, in higher KLK6 expression, which enhances the potential role of the protein as a biomarker (*Schrader et al., 2015*), which we wanted to examine in the study.

## MATERIALS AND METHODS

### Patient cohort

The study protocol was performed in accordance with institutional bioethics guidelines and was approved by the Bioethics Committee of Medical University of Warsaw (approval number: KB/71/2018). All patients provided written informed consent. All procedures performed in the study were in accordance with the ethical standards of the national research committee and with the 1964 Helsinki declaration and its later amendments.

This prospective study included 80 consecutive patients with precancerous and cancerous vocal fold lesions treated at the single otorhinolaryngology center. The patient cohorts included 55 (68.75%) men and 25 (31.25%) women, aged from 24 to 90 years (mean age 65.1 ± 12.87). All included patients had a primary clinical diagnosis of

**Table 1 Characteristics of patients.**

| Patient cohort characteristics | Values |
| --- | --- |
| Mean age | 65.1 ± 12.87 |
| Gender | |
| Female | 25 (31.25%) |
| Male | 55 (68.75%) |
| Histopathology | |
| Non-dysplasia | 39 (48.75%) |
| Low grade dysplasia | 8 (10%) |
| High grade dysplasia | 6 (7.5%) |
| Invasive cancer | 27 (33.75%) |
| Smoking status | |
| Non-smokers | 27 (34.2%) |
| Smokers | 52 (65.8%) |
| GERD symptoms | |
| No | 65 (82.3%) |
| Yes | 14 (17.7%) |

Note:
GERD, gastroesophageal reflux disease.

hypertrophic vocal fold lesion without impairment of vocal fold mobility. We excluded from analysis those, who had a history of systemic immunotherapy, chemotherapy, diagnosis of other neoplasms. All participants had surgical resection of the lesion with $CO_2$ laser transoral laryngeal microsurgery. Blood samples were collected from each patient on the day of surgery, but before the resection. Final diagnosis was confirmed on histopathological examination and included 39 (48.75%) non dysplastic lesions, eight (10%) low-grade dysplasia, six (7.5%) high-grade dysplasia and 27 (33.75%) invasive cancers. From medical records the data involving the presence of gastroesophageal reflux disease (GERD) symptoms and smoking status were extracted (Table 1).

## Enzyme-linked immunosorbent assay (ELISA)

Patients' blood samples were centrifuged at 4 °C to retrieve the serum. Serum of each patient was aliquoted in four samples, and stored at −80 °C in the biological resources repository at the department of laboratory diagnostics. National ethical guidelines of human tissues collection were preserved. The serum samples were thawed shortly before determination of biomarkers expression by enzyme-linked immunosorbent assay (ELISA). The ELISA procedures were performed according to the manufacturer's instruction. Individual serum concentration of selected proteins were reported in ng/mL.

The following sandwich enzyme immunoassays were performed:

- Human Vascular Endothelial-Cadherin Complex (VE-cad) (Sunred–Shanghai Sunred Biological Technology Co., Ltd; China; catalogue number: 201-12-0273; range: 0.15–40 ng/mL)
- Human CD44 ELISA Kit (EIAab–Wuhan EIAab Science Co., Ltd; China; catalogue number: E0670h; range: 0.156–10 ng/mL)

- Human High mobility group protein B1(HMGB1) ELISA Kit (Sunred–Shanghai Sunred Biological Technology Co., Ltd; China; catalogue number: 201-12-1636; range: 2–200 ng/mL)
- Human Kallikrein 6 (KLK 6) ELISA Kit (Sunred–Shanghai Sunred Biological Technology Co., Ltd; China; catalogue number: 201-12-0863; range: 0.25–70 ng/mL)

## Statistical analysis

Statistical analysis was performed with IBM SPSS Statistics 26.0. and Statistica 13.3. The Mann–Whitney U tests and Kruskal–Wallis tests were used to analyze the statistical associations of biomarkers expression levels with clinicopathological parameters. The correlation between measures were evaluated using Spearman's rank correlation coefficients. $p$-values lower than 0.05 were considered to indicate a statistically significant results.

## RESULTS

The highest mean levels of HMGB1, KLK6 and VE-cad were detected in sera of patients with low-grade dysplasia (81.14, 24.33, 14.17 respectively) but soluble CD44 was the most elevated in patients with non-dysplastic lesions (2.49). There was observed differentiation of analysed biomarkers serum levels among histopathological stages with the observed trend for HMGB1, KLK6 and VE-cad of increasing concentration from non-dysplastic to low-grade dysplasia and followed by the decrease for high-grade dysplasia and invasive cancer, however the differences were not significant ($p$-values 0.897, 0.354, 0.1 respectively). The patients' serum had the highest CD44 concentration in non-dysplastic and low-grade dysplasia with the following decrease through high-grade dysplasia and invasive cancer. Analyzes of the predictive potential of groups of proteins were also performed by combining them into pairs, threes and all four together. Unfortunately, the combined results were also not correlated to the stage of laryngeal lesions (Fig. 1).

In 14 patients with GERD symptoms, the levels of KLK6 and CD44 were higher than in patients without them and the differences were close to statistical significance ($p$-value 0.06 and 0.084 respectively). There were no significant differences of biomarkers levels related to patients gender ($p$-value from 0.243 to 1) or smoking status ($p$-value from 0.22 to 0.706). Exact test statistics, degrees of freedom and size effects are presented in the Table S2, Table 2.

The serum expression levels of HMGB-1, KLK6 and VE-cad showed significant correlations between each of the three biomarkers (range from 0.553 to 0.681). Contrary the level of CD44 was not correlated with other biomarkers. None of the biomarkers was statistically correlated with patients age (Table 3).

## DISCUSSION

The appearance of the hypertrophic laryngeal lesions do not often correspond to their histopathological advancement (*Gale et al., 2016*; *Gale, Poljak & Zidar, 2017*; *Schaaij-Visser et al., 2010*). Therefore, the differential diagnosis may result in delaying the final diagnosis and initiation of the therapy. Currently, the tissue histopathology still remains

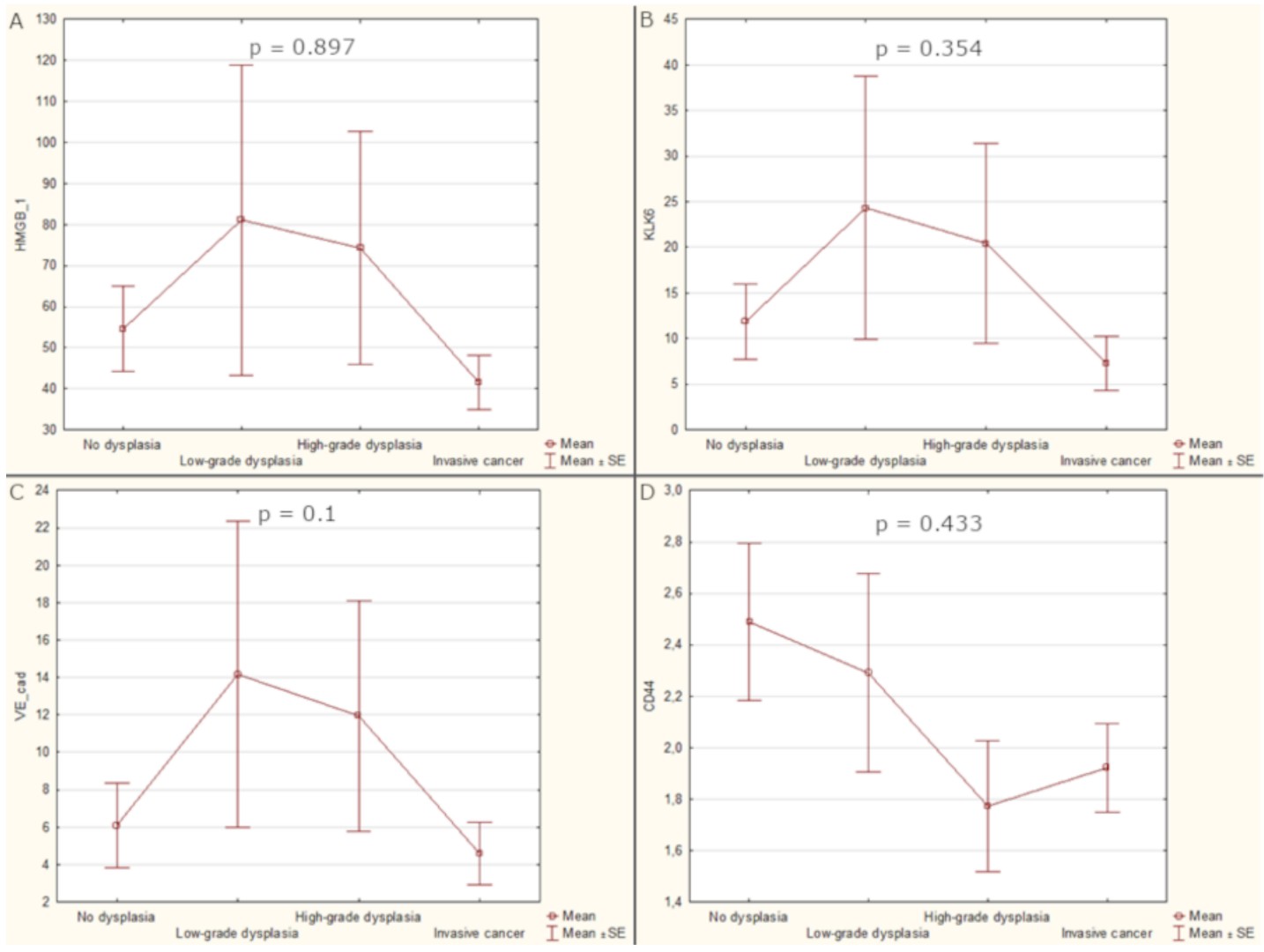

**Figure 1 Average values of biomarker levels in patients' sera with standard errors (SE).** (A) Serum level of HGMB1. (B) Serum level of KLK6. (C) Serum level of VE-cad. (D) Serum level of CD44.

the standard, however the accurate and efficient biomarker, available in preoperative setting could optimize the therapeutic decisions in laryngeal intraepithelial lesions.

In our study four potential biomarkers (VE-cad, HMGB1, CD44 and KLK6) were evaluated. According to the literature, all of them seemed to be promising proteins in facilitating the cancer diagnosis and potentially useful for differentiation between non-dysplastic lesions and invasive cancer. Although the presented mean levels were not significantly different among stages of intraepithelial lesions and invasive cancer, there were recognized specific patterns of concentration for HMGB1, KLK6 and VE-cad different from those for CD44 levels. Interesting and requiring further researches is also the higher serum expression of KLK6 and CD44 in patients with GERD symptoms that were close to significance ($p = 0.06$ and $p = 0.084$).

**Table 2 Relationships between serum level expression of biomarkers and patients clinicopathological factors.**

| Variable | HMGB1 | | KLK6 | | VE-cadherin | | CD44 | |
|---|---|---|---|---|---|---|---|---|
| | Mean ± SD | p-value | Mean ± SD | p-value | Mean ± SD | p-value | Mean ± SD | p-value |
| Gender | | | | | | | | |
| Female | 69.55 ± 67.8 | 0.243* | 17.61 ± 27.06 | 0.493* | 9.35 ± 15.24 | 1* | 2.11 ± 1.4 | 0.51* |
| Male | 44.24 ± 47.12 | | 8.59 ± 19.81 | | 5.27 ± 11.19 | | 2.13 ± 1.19 | |
| Histopathology | | | | | | | | |
| Invasive cancer | 41.52 ± 40.98 | 0.897# | 7.28 ± 18.49 | 0.354# | 4.58 ± 10.47 | 0.1# | 1.92 ± 1.08 | 0.433# |
| High grade dysplasia | 74.27 ± 80.29 | | 20.43 ± 31 | | 11.95 ± 17.42 | | 1.77 ± 0.72 | |
| Low grade dysplasia | 81.14 ± 92.64 | | 24.33 ± 35.39 | | 14.17 ± 20.01 | | 2.29 ± 0.94 | |
| No dysplasia | 54.5 ± 54.03 | | 11.82 ± 21.34 | | 6.1 ± 11.76 | | 2.49 ± 1.58 | |
| Smoking | | | | | | | | |
| Non-smokers | 49.31 ± 53.7 | 0.645* | 9.84 ± 20.24 | 0.405* | 5.97 ± 11.59 | 0.22* | 2.3 ± 1.49 | 0.706* |
| Smokers | 54.1 ± 56.92 | | 12.42 ± 23.96 | | 6.96 ± 13.35 | | 2.04 ± 1.13 | |
| GERD | | | | | | | | |
| No | 50.27 ± 54.82 | 0.23* | 10.45 ± 22.1 | 0.06* | 6.51 ± 12.78 | 0.418* | 2.22 ± 1.29 | 0.084* |
| Yes | 62.27 ± 59.82 | | 16.61 ± 25.33 | | 7.14 ± 12.8 | | 1.72 ± 1.05 | |

**Note:**
* Mann–Whitney U test.
# Kruskal–Wallis test.

**Table 3 Spearman's rank correlation coefficients between biomarkers' serum levels.**

| Correlations | HMGB1 | KLK6 | VE-cadherin | CD44 |
|---|---|---|---|---|
| HMGB1 | | | | |
| $r_s$ (effect size) | – | 0.681* | 0.553* | 0.17 |
| n | | 80 | 80 | 80 |
| p-value | | 3.645E−12 | 1.057E−7 | 0.131 |
| KLK6 | | | | |
| $r_s$ (effect size) | 0.681* | – | 0.623* | 0.018 |
| n | 80 | | 80 | 80 |
| p-value | 3.645E−12 | | 6.543E−10 | 0.874 |
| VE-cadherin | | | | |
| $r_s$ (effect size) | 0.553* | 0.623* | – | −0.031 |
| n | 80 | 80 | | 80 |
| p-value | 1.057E−7 | 6.543E−10 | | 0.784 |
| CD44 | | | | |
| $r_s$ (effect size) | 0.17 | 0.018 | −0.031 | – |
| n | 80 | 80 | 80 | |
| p-value | 0.131 | 0.874 | 0.784 | |

**Note:**
* p-value lower than 0.05.

So far there have been no studies evaluating VE-cadherin serum level in patients with laryngeal dysplasia or cancer. Only a few studies have explored soluble VE-cadherin in several types of human solid cancers. In *Sulkowska et al. (2006)*, the level of VE-cadherin

was significantly higher in sera of patients with colorectal cancer in comparison to healthy controls (1.68 ng/mL vs 0.42 ng/mL; $p < 0.00001$). However, *Allegrini et al. (2012)* did not show significant differences in serum expression level of VE-cadherin between patients with and without refractory gastrointestinal cancers. *Irani & Dehghan (2018)* evaluated the expression of VE-cadherin in oral squamous cell carcinoma tissues and found statistically significant differences between tumor grade and the expression levels of VE-cadherin ($p < 0.001$) and the expression was higher in high grade cancers. This is the only study exploring VE-cadherin in head and neck cancers (HNC); moreover, it analyzed the tissue expression and not the serum.

In regard to the level of HMGB1, the study by *Wild et al. (2012)* revealed significantly higher level of the marker in patients with HNC *versus* healthy donors ($p = 0.002$). Similar results were published by *Qiu et al. (2014)*, who confirmed significantly higher level of serum HMGB1 in patients with laryngeal cancer compared with the healthy controls (4.81 *vs.* 3.21 ng/mL, $p < 0.001$). Also, the usefulness of the serum HMGB1 as a potential biomarker was evaluated and the cut-off value of serum HMGB1 level was 4.80 ng/ml, the sensitivity and specificity in cancer identification were respectively 42.3% and 92.0%. However, we did not find any works evaluating the usefulness of this biomarker in laryngeal dysplastic lesions. Surprisingly, in our results, the highest level of serum HMGB1 was identified in patients with low-grade dysplasia.

The importance of CD44 as a biomarker in HNC has been proven many times, as summarized in the meta-analysis by *Chen et al. (2014)*. However, this meta-analysis included only studies exploring immunohistochemical staining, not plasma expression, and most of the available studies have compared advanced-stage head and neck tumors without inclusion of patients with dysplastic lesions. Similar results as in our study were obtained by *Seyedmajidi et al. (2018)*. The study included 20 patients with oral squamous cell carcinoma and 20 healthy volunteers. There was no statistically significant difference in serum CD44 level between the patient and control groups ($p$-value = 0.182). Attention should be paid also to the specificity of the CD44 molecule. As noted in the introduction, the CD44 molecule is structured differently depending on the type and order of exon splicing. In *Shinohara et al. (2016)*, the serum level of CD44 was higher in male patients and smokers *versus* females ($p = 0.006$) and non-smokers ($p = 0.001$). The study focused on the clinical significance of the expression of CD44 in non-small cell lung cancer (NSCLC). In the cited study, the stated differences in expression were only shown for CD44v6, but not for CD44st, so the outcomes depends on the specific structure of the molecule. In our study we checked expression of CD44 generally, which could influence the study's results.

According to the study by *Zhang et al. (2014)*, the plasma level of KLK6 was significantly higher in the LSCC patients (median, interquartile range: 4.90, 4.26) than in the benign lesions (median, interquartile range: 4.65, 1.86; $p = 0.047$) and healthy cohort (median, interquartile range: 4.65, 2.32; $p = 0.014$). There was no significant correlation between the plasma KLK6 levels and clinicopathological characteristics. There are no other studies verifying the usefulness of KLK6 in the diagnosis of LSCC. However, *Tailor et al. (2018)* proved that gene KLK6 expression was decreased in HNC compared to healthy tissues.

The protease form of the kallikrein family–KLK8–was tested in *Linkov et al. (2007)* and the results were opposite to the findings of *Zhang et al. (2014)*. The median level of KLK8 was shown to be statistically lower between the active HNC and the healthy control group ($p < 0.001$).

The limitation of the presented study is the relatively small cohort of patients. The lack of statistical significance of the average biomarker levels may result from the insufficient numbers of individual diagnoses. Moreover, we aimed the evaluation of the biomarkers in different stages of early intraepithelial laryngeal lesions and did not include patients with advanced cancer stages and healthy participants. Most of the studies that have been published so far compared the serum biomarkers levels in patients with advanced cancer and healthy controls (*Allegrini et al., 2012*; *Chen et al., 2014*; *Seyedmajidi et al., 2018*; *Sulkowska et al., 2006*; *Zhang et al., 2014*).

Another reason for the lack of significant differences in the mean serum concentrations of biomarkers may be the choice of reagents. Depending on the manufacturer, the ranges of reagents for ELISA tests differ significantly. Different range and quality of reagents certainly have an influence on the obtained results.

## CONCLUSIONS

VE-cad, HMGB1, CD44 and KLK6 did not prove to be reliable biomarkers that implicate malignant potential within vocal fold hypertrophic intraepithelial lesions.

### Funding
The study was supported by the internal grant of Medical University of Warsaw (1WF/NM1/18). The funders had no role in study design, data collection and analysis, decision to publish, or preparation of the manuscript.

### Grant Disclosures
The following grant information was disclosed by the authors:
Medical University of Warsaw: 1WF/NM1/18.

### Competing Interests
The authors declare that they have no competing interests.

### Author Contributions
- Michał Żurek conceived and designed the experiments, performed the experiments, analyzed the data, prepared figures and/or tables, authored or reviewed drafts of the paper, and approved the final draft.
- Anna Rzepakowska conceived and designed the experiments, analyzed the data, prepared figures and/or tables, and authored or reviewed drafts of the paper approved the final draft.
- Iwona Kotuła conceived and designed the experiments, performed the experiments, prepared figures and/or tables, and approved the final draft.

- Urszula Demkow conceived and designed the experiments, and approved the final draft.
- Kazimierz Niemczyk conceived and designed the experiments, and approved the final draft.

## Human Ethics

The following information was supplied relating to ethical approvals (*i.e.*, approving body and any reference numbers):

The study was conducted according to the guidelines of the Declaration of Helsinki and approved by the Bioethics Committee of Medical University of Warsaw (KB/71/2018). Informed written consents were obtained from all patients.

## Data Availability

The raw measurements are available in the Supplemental File.

## Supplemental Information

Supplemental information for this article can be found online at http://dx.doi.org/10.7717/peerj.13104#supplemental-information.

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
