# Peer review of "Serum expression of Vascular Endothelial-Cadherin, CD44, Human High mobility group B1, Kallikrein 6 proteins in different stages of laryngeal intraepithelial lesions and early glottis cancer"

_PeerJ, doi:10.7717/peerj.13104_

## Round 0.1 · original submission · Major Revisions

Thank you for your submission. You are invited to address all comments by the reviewers and resubmit a revised manuscript with detailed description of changes made.

Reviewer 1 ·

Basic reporting

The rationale for choosing the markers is not sufficiently described in the manuscript. The introdcution should include more about the markers, and their function, along with a brief, but clear background on why the authors chose these markers.

Throughout the manuscript, there are sentences that are not clear and should be re-worded to ensure that an international audience can clearly understand the text. Some examples include lines 58-59, 78-80.

Experimental design

The findings of the study can be made comprehensive by performing in-silico analysis on transcriptomic data to determine the importance of the specific markers in LSCC vs normal tissue.

Validity of the findings

No comment

Additional comments

The research study by Żurek et al describes the validation of molecular markers- Vascular Endothelial-Cadherin Complex (VE-cad), CD44, Human High mobility group protein B1(HMGB1), and Kallikrein 6 in serum samples from patients with specific stages of laryngeal intraepithelial lesions to serve as diagnostic tools in differentiation of benign and dysplastic lesions from invasive pathologies. The research study is novel, and the authors provide a strong rationale for carrying out the study to identify novel biomarkers preoperatively in serum or saliva to guide therapeutic decisions in laryngeal squamous cell carcinoma. The authors have comprehensively discussed the pitfalls and shortcomings of the study and methods. The authors found the markers under study to not be reliable biomarkers implicating malignant potential within vocal fold hypertrophic intraepithelial lesions.

·

Basic reporting

The manuscript entitled “Serum expression of Vascular Endothelial-Cadherin, CD44, Human High mobility group B1, Kallikrein 6 proteins in different stages of laryngeal intraepithelial lesions and early glottis cancer” by Michał Żurek et al. investigated the potential value of VE-cad, HMGB1, CD44 and KLK6 in serum samples from patients with non-dysplastic lesions, low-grade dysplasia, high-grade dysplasia and invasive cancers as diagnostic markers. They have found that the four proteins are not associated with the stage of laryngeal lesions.

Experimental design

VE-cad, HMGB1, CD44 and KLK6 were not related to disease stages in the human cohort in this manuscript. Authors could devise a diagnostic signature including these four proteins and measure whether the expression of the 4-gene signature is correlated to the stage of laryngeal lesions.

Validity of the findings

Are the four markers, VE-cad, HMGB1, CD44 and KLK6 associated with survival and disease stage of patients with laryngeal cancers? Are they differentially expressed in tumor compared to normal tissue? To validate the findings in this manuscript, authors could investigate in published human cohorts, like TCGA and METABRIC.

Additional comments

None.

Reviewer 3 ·

Basic reporting

In the discussion part, for each biomarker, the author compare their expression levels between different cancers, which is not very scientific.

Experimental design

1. No legend for all figure and tables
2. Please label p value on figure1.

Validity of the findings

3. Please point out the clinical meaning of this study, which was not spelled out very clearly.

---

## Round 0.2 · accepted · Accept

Your revised manuscript was reviewed again by the reviewers. They appreciated your revisions and found them satisfactory. Thank you for addressing them.

·

Basic reporting

Authors have successfully responded to my comments.

Experimental design

None.

Validity of the findings

None.

Additional comments

None.

Reviewer 3 ·

Basic reporting

Thanks for the changes made by the authors point by point.

Experimental design

Good.

Validity of the findings

Good

Additional comments

No.